Accepted at the ICLR 2024 Workshop on AI4Differential Equations In Science

# NEURAL OPERATORS WITH LOCALIZED INTEGRAL AND DIFFERENTIAL KERNELS

**Miguel Liu-Schiaffini**,* **Julius Berner**\* **& Anima Anandkumar**
California Institute of Technology
{mliuschi, jberner, anima}@caltech.edu

**Boris Bonev**,* **Thorsten Kurth & Kamyar Azizzadenesheli**
NVIDIA
{bbonev, tkurth, kamyara}@nvidia.com

## ABSTRACT

Neural operators learn mappings between function spaces, which is applicable for learning solution operators of PDEs and other scientific modeling applications. Among them, the Fourier neural operator (FNO) is a popular architecture that performs global convolutions in the Fourier space. However, such global operations are often prone to over-smoothing and may fail to capture local details. In contrast, convolutional neural networks (CNN) can capture local features but are limited to training and inference at a single resolution. In this work, we present a principled approach to operator learning that can capture local features under two frameworks by learning differential operators and integral operators with locally supported kernels. Specifically, inspired by stencil methods, we prove that under an appropriate scaling of the kernel values of CNNs, we obtain differential operators. To obtain integral local operators, we utilize suitable basis representations for the kernels based on discrete-continuous convolutions. Both these principled approaches preserve the properties of operator learning and, hence, the ability to predict at any resolution. Adding our layers to FNOs significantly improves their performance, reducing the relative $L^2$-error by 34-72% in our experiments on turbulent 2D Navier-Stokes fluid flow and the spherical shallow water equations.

## 1 INTRODUCTION

Deep learning holds the promise to greatly accelerate advances in computational science and engineering, which often require numerical solutions of partial differential equations (PDEs) (Azzizadenesheli et al., 2023; Zhang et al., 2023; Cuomo et al., 2022). Large speedups over traditional methods have been achieved by the usage of neural operators, which learn mappings between function spaces, enabling operator learning for function-valued data (Li et al., 2021; Azzizadenesheli et al., 2023; Raonić et al., 2023). In particular, they are agnostic to the discretization of the input and output functions—a vital property in the context of PDEs where data is often provided at varying resolutions and high-resolution data is costly to generate (Kovachki et al., 2021). In contrast, standard neural networks such as convolutional neural networks (CNN) (Ronneberger et al., 2015; Gupta & Brandstetter, 2022) require the functions to be discretized at a fixed resolution on a regular grid, which is limiting. In the past few years, various architectures of neural operators have been developed. Among them, the Fourier neural operator (FNO) (Li et al., 2020a) has gained popularity and shown good performance in a number of applications. However, FNO performs global convolutions in the Fourier space, which is often prone to over-smoothing and may fail to capture local details.

There are many applications that require a local neural operator. For instance, solution operators of several relevant PDEs are of local nature. Examples include hyperbolic PDEs, which have real-valued characteristic curves (LeVeque, 1992). As a result, a solution at a given point will only depend on the initial condition within a neighborhood of that point. As such, their solution operators only

---
*Equal contribution

Table 1: Comparison of different architectures for the solution of PDEs. The top half enumerates architectures for planar domains and the bottom half for spherical domains. Our proposed architectures are highlighted in bold letters, and the differential and integral kernels are detailed in Section 2.

| Architecture | Efficient | Receptive field | no input downsampling |
|---|---|---|---|
| GNO | ✗ | local/global | ✓ |
| FNO | ✓ | global | ✓ |
| CNO / U-Net | ✓ | local | ✗ |
| **FNO + integral** | ✓ | local/global | ✓ |
| **FNO + differential** | ✓ | local/global | ✓ |
| SFNO | ✓ | global | ✓ |
| **SFNO + integral** | ✓ | local/global | ✓ |

have a local receptive field and can, therefore, be efficiently learned by locally supported kernels. Further examples of local operators are differential operators, which can be expressed in terms of pointwise multiplication with the frequency in the spectral domain. Consequently, they introduce large errors when approximated by a finite number of parameters in Fourier space. In this context, we note that the emulation of classical numerical methods to solve partial differential equations, such as finite-difference methods, relies on the usage of local stencils for differentiation. This calls for the presence of local and differential operators in neural operator architectures.

Naturally, every local operation can also be represented by a global operation. However, this is typically vastly parameter-inefficient and does not provide a good inductive bias for learning local operations. In the context of neural operators, spectral variants of neural operators, such as the FNO and Spherical FNO (SFNO) (Bonev et al., 2023), are theoretically able to approximate local convolutions. However, representing local kernels requires the approximation of a global signal in the spectral domain, in turn demanding a large number of parameters (due to the uncertainty principle).

A few prior works have explored the usage of local operations in the context of neural operators (see Appendix A). However, since all of the above approaches rely on standard convolutional layers on equidistant grids, they have the following shortcomings. First, such approaches do not allow for a natural extension to unstructured grids or other geometries, which are ubiquitous in PDE problems (Li et al., 2023). Moreover, they can only be applied to higher resolutions by downsampling of the (intermediate) inputs to the training resolution. Important high-frequency content can be lost through downsampling, which is particularly problematic for multi-scale data in the context of PDEs. In contrast, we develop convolutional layers that can be applied at any resolution without downsampling. We achieve this by appropriately scaling the receptive field or the values of the kernel (see also Table 1 and Figure 1).

## 2 LOCAL LAYERS

In this section, we develop computationally efficient and principled approaches to include operations in neural operators that capture local receptive fields, while retaining the ability to approximate operators and hence, extend to multiple resolutions. We consider two kinds of localized operators: differential operators and integral kernel operators with a locally supported kernel (see Figure 1). For the first, we draw inspiration from stencils of finite-difference methods. We derive conditions to modify convolutional layers such that they converge to a unique differential operator when the discretization is refined. For the second case of local integral operator, we adapt discrete-continuous (DISCO) convolutions (Ocampo et al., 2022) to provide an efficient, discretization-agnostic framework that can be applied to general meshes on both planar and spherical geometries.

We consider an input function $v \colon \mathbb{R}^d \supset D \to \mathbb{R}^n$ that is discretized on meshes $D^h \subset D$ with width $h$ on a domain $D \subset \mathbb{R}^d$. For notational convenience, we present most ideas for the case $d = 1$, but it is straightforward to extend them to higher-dimensional domains.

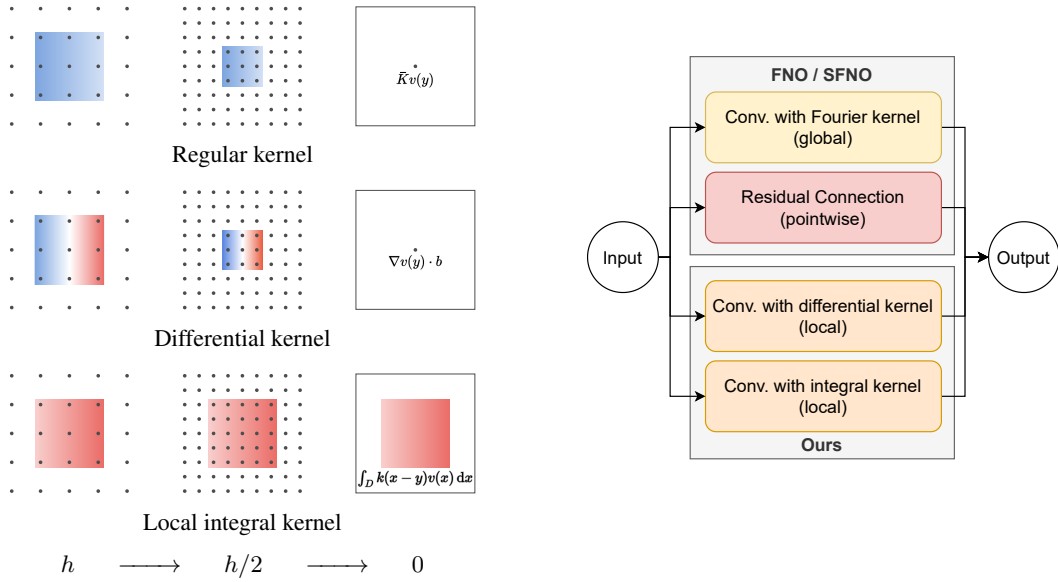

Figure 1: (**Left**): Visualization of different limits of a convolution with a discretized function $v$ as the grid width $h$ is refined, i.e., $h \to 0$. A regular convolution is collapsing to a pointwise linear operator (top), whereas a differential operator or a local integral operator can be achieved by correctly constraining the kernel size and weights in the limit, see Section 2. (**Right**) A single layer of our local neural operator. We add (up to) two local operations using the convolutions with differential kernel and local integral kernel (Section 2).

**Differential Layer:** We construct a layer that converges to a differential operator when the width $h$ of the discretization $D^h$ tends to zero. To prevent the operator from collapsing to a pointwise operation, we constrain and rescale the values of the kernel (according to the discretization width). Note that without rescaling, we would again recover a pointwise operator; however, due to the rescaling by a factor of $\frac{1}{h}$, the limit might not exist without constraining the values of the kernel. The next proposition shows that it is sufficient to subtract the average kernel value. See Appendix B for a more general statement and a corresponding proof.

**Proposition 2.1** (First-order differential layer). *Let $D_h \subset \mathbb{R}^d$ be a regular grid of width $h$ and let $v \in C^1(D, \mathbb{R}^n)$. Then, for every kernel $(K_i)_{i=1}^S \subset \mathbb{R}^n$, there exists $(b_j)_{j=1}^n \subset \mathbb{R}^d$ such that*

$$\lim_{h \to 0} \frac{1}{h} \text{Conv}_{K-\bar{K}}[v](y) = \sum_{j=1}^n \nabla v_j(y) \cdot b_j$$

*for every $y \in D_h$, where $\bar{K} = \sum_{i=1}^S K_i$.*

Proposition 2.1 shows that we can learn different directional derivatives using an appropriate adaptation of standard convolutional layers. Specifically, we center the kernel $K$ by subtracting its mean $\bar{K}$ and scale the result by the reciprocal resolution $\frac{1}{h}$.

**Integral Kernel Layers:** Instead of rescaling the kernel, we can also adapt the size of the kernel such that the receptive field stays the same, i.e., is independent of the resolution $h$. We will explain in the following how we can adapt the size of the kernel with a fixed number of parameters. To construct a locally supported kernel $\kappa$, we can directly discretize a convolution, i.e.,

$$\int_D k(x, y) \cdot v(x) \, dx \approx \sum_{x \in D^h} \kappa(x - y) \cdot v(x) \, q_x. \tag{1}$$

Note that the sum can be taken only over the $x \in D^h$ with $x - y \in \text{supp}(\kappa)$.

The remaining question is centered around the parametrization of the kernel. One could draw inspiration from works on *hypernetworks* in computer vision and parameterize it using a neural

Table 2: Experimental results for Darcy flow, Navier-Stokes, and the spherical shallow water problems. For all three problems, the test error is reported in terms of the relative $L^2$-loss after a single step. For the time-dependent Navier-Stokes and shallow water equations we also predict the error after 5 autoregressive steps.

| PDE | Model | rel. $L^2$-error (1 step) | rel. $L^2$-error (5 steps) |
|---|---|---|---|
| Darcy Flow | U-Net | $1.380 \cdot 10^{-2}$ | - |
| | FNO | $5.867 \cdot 10^{-2}$ | - |
| | **FNO + diff. kernel (ours)** | $\mathbf{7.357 \cdot 10^{-3}}$ | - |
| | FNO + local integral kernel (ours) | $6.034 \cdot 10^{-2}$ | - |
| | FNO + local integral + diff. kernel (ours) | $9.032 \cdot 10^{-3}$ | - |
| Navier-Stokes | U-Net | $1.674 \cdot 10^{-1}$ | $5.115 \cdot 10^{-1}$ |
| | FNO | $1.381 \cdot 10^{-1}$ | $2.360 \cdot 10^{-1}$ |
| | FNO + diff. kernel (ours) | $1.073 \cdot 10^{-1}$ | $2.129 \cdot 10^{-1}$ |
| | FNO + local integral kernel (ours) | $1.110 \cdot 10^{-1}$ | $2.183 \cdot 10^{-1}$ |
| | **FNO + local integral + diff. kernel (ours)** | $\mathbf{9.022 \cdot 10^{-2}}$ | $\mathbf{1.956 \cdot 10^{-1}}$ |
| Shallow Water | U-Net | $1.341 \cdot 10^{-3}$ | $1.226 \cdot 10^{-2}$ |
| | Spherical U-Net (with local integral kernel) | $6.160 \cdot 10^{-4}$ | $3.265 \cdot 10^{-3}$ |
| | SFNO | $9.220 \cdot 10^{-4}$ | $3.185 \cdot 10^{-3}$ |
| | **SFNO + local integral kernel (ours)** | $\mathbf{2.624 \cdot 10^{-4}}$ | $\mathbf{5.392 \cdot 10^{-4}}$ |

network (Shocher et al., 2020). However, this is significantly more costly than a standard convolution. Another idea is to interpolate a fixed-sized kernel, e.g., using sinc or bi-linear interpolation. A more general version of the latter, based on a learnable linear combination of hat functions, is also known as *discrete-continuous* (DISCO) convolution and has shown to be effective in different applications (Ocampo et al., 2022). We note that this formulation is closest to the original convolutional layer typically used in computer vision. However, in Appendix C we see that it also allows us to use unstructured meshes and formulate the operation on more general domains $D \subset \mathbb{R}^d$.

**Local neural operator architecture:**  To design our neural operator, we want to combine pointwise, local, and global operations. To this end, we take a FNO (or SFNO for spherical problems) as a starting point, which already features global operations in Fourier space and pointwise operations using its residual connections. Then, we augment the operators by adding our proposed local convolutions from the previous section as additional branches in the respective layers (see Figure 1).

## 3  EXPERIMENTS

To validate the effectiveness of local operators, we evaluate the architecture on three PDE problems. Figure 2 shows samples on two of these PDEs. In all three cases, we incorporate our proposed local layers into existing FNO and SFNO architectures to demonstrate that a significant improvement in performance can be achieved by introducing the inductive bias of local convolutions (see Figure 1 and Section 2 for details on the architecture). We focus on three experimental settings: learning a differential operator in a Darcy flow problem, a 2D Navier-Stokes fluid flow with Reynolds number 5000, and the spherical shallow water equations, see Appendix D. The detailed experimental setting, specific implementation details, and hyperparameters can be found in Appendix E.

The results of our numerical experiments are reported[1] in Table 2. We observe significant performance gains over the baselines in all three problem settings. In particular, we notice that the best performance in the reported relative $L^2$-errors is achieved with the inclusion of both global and local convolutions[2], despite an overall reduction in parameter count with respect to their corresponding FNO/SFNO baselines. This is particularly pronounced for the Navier-Stokes and shallow water equations, where

---

[1]In all three settings, hyperparameters are chosen to result in similar macro-architectures with roughly similar parameter counts for the purpose of comparability. The experimental setup including choice of hyperparameters is outlined in Section E in the Appendix.

[2]Similar results are also observed in the setting of zero-shot super-resolution, which are reported in Appendix E.4.

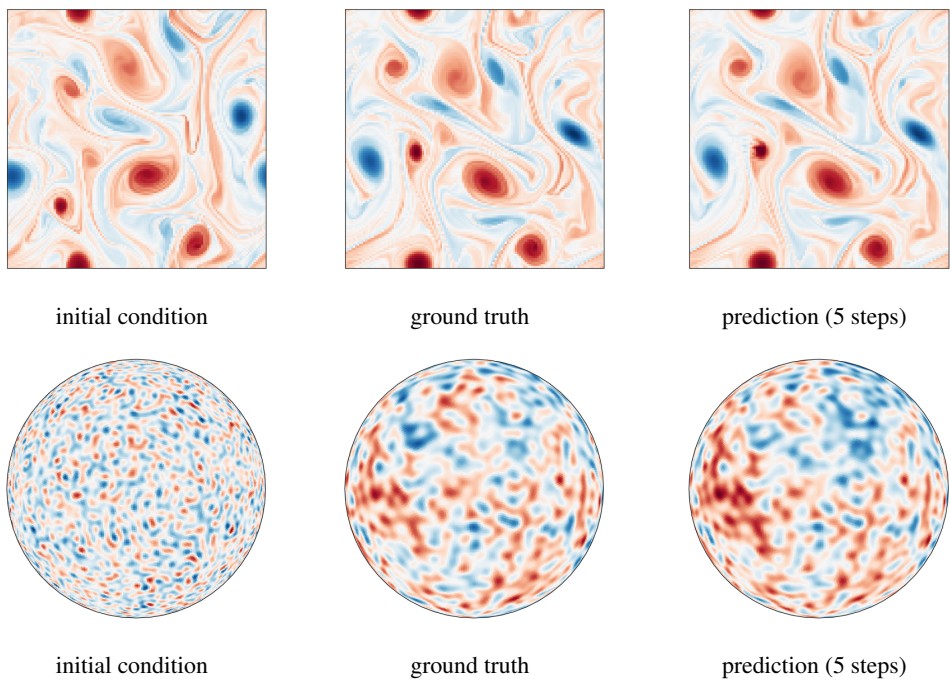

initial condition        ground truth        prediction (5 steps)

initial condition        ground truth        prediction (5 steps)

Figure 2: Initial condition, ground truth, and corresponding autoregressive predictions of our proposed models for the Navier-Stokes problem (top row) and the shallow water equations (bottom row).

errors can accumulate in autoregressive roll-outs. We attribute this to our models' inductive bias, allowing it to better capture the fine-grained scales and thus achieve better performance.

The Navier-Stokes problem demonstrates the efficacy of combining both differential and local integral kernels with the global convolution of the FNO. Our proposed model with these three components together outperforms models with only two of these three operations. We note that the Darcy flow problem is meant to motivate the need for our proposed differential kernels (and indeed, our best-performing model achieves $87\%$ lower relative $L^2$-error than FNO). Since the ground truth operator in our Darcy problem is a differential operator, we expect (and observe) that for high accuracies, very local (i.e., differential) kernels are needed. In particular, we see that the baseline FNO performs poorly and the local integral kernels do not provide additional benefit. However, we include them in Table 2 for completeness.

## 4 CONCLUSION

In this paper, we have demonstrated a novel framework for local neural operators. We have shown how convolutional layers can be constrained to realize neural operators that approximate differential operators in the continuous limit. Moreover, we have derived convolutions with local integral kernels from the general notion of an integral transform and the related graph neural operator. Finally, we have constructed localized neural operators on the sphere by using discrete-continuous convolutions Ocampo et al. (2022).

The resulting neural operators introduce a strong inductive bias for learning operators with local receptive fields. In particular, their formulation ensures the same local operation everywhere in the domain. This equivariance (w.r.t. the underlying symmetry group) reduces the required number of learnable parameters and improves generalization.

Our numerical experiments demonstrate consistent improvements when existing neural operators with global receptive fields are augmented with the proposed localized convolutions, resulting in reductions in relative $L^2$-error of up to 72% over the corresponding baselines. We thus expect local neural operators to play an important role in solving real-world scientific computing problems with machine learning.

ACKNOWLEDGMENTS

M. Liu-Schiaffini is grateful for support from the Mellon Mays Undergraduate Fellowship. J. Berner acknowledges support from the Wally Baer and Jeri Weiss Postdoctoral Fellowship. A. Anandkumar is supported in part by Bren endowed chair and by the AI2050 senior fellow program at Schmidt Sciences.

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

## A    RELATED WORK AND CONNECTIONS TO OTHER FRAMEWORKS

We strive to formulate local operations that are consistent with the neural operator paradigm, which stipulates that operations within the model are independent of the discretization of input functions (Kovachki et al., 2021). In principle, this allows the evaluation of the neural operator on arbitrary meshes[3].

A few prior works have explored the usage of local operations in the context of neural operators. Li et al. (2020b) introduce graph neural operators (GNOs), which parameterize local integral kernels with a neural network. However, evaluating this network on all combinations of points to compute the integrals can make GNOs computationally expensive and slower in practice than hardware-optimized convolutional kernels. Alternatively, Ye et al. (2022; 2023) propose local neural operators by combining FNOs with convolutional layers and, similarly, Wen et al. (2022) integrate U-Nets and FNOs. Moreover, Raonić et al. (2023) propose convolutional neural operators (CNOs) by leveraging U-Nets that (approximately) respect the band-limits.

Several other discretization-independent local operations have been introduced in the deep learning literature. While some of these approaches have not detailed the connections to neural operators, they are independent of the discretization. We outline the connections between these works in the following:

**GNOs and Hypernetworks:**    In general, local operations can be provided by graph neural operators (GNOs) (Li et al., 2020b). The GNO is given by

$$\mathrm{GNO}_k[v](y) = \int_{U(y)} k(x, y) \cdot v(x)\, \mathrm{d}x \approx \sum_{x \in D^h \cap U(y)} k(x, y) \cdot v(x)\, q_x, \tag{2}$$

where $k$ is a kernel (typically parametrized by a neural network) and $q_x$ are suitable quadrature weights. While the GNO can represent local kernel integral operators by picking a suitably small neighborhood $U(y) \subset D$, the evaluation of the kernel and aggregation in each neighborhood $U(y)$ is slow and memory-intensive for general kernels $k \colon D \times D \to \mathbb{R}^n$ (Li et al., 2020c). Moreover, for an arbitrary kernel, GNO is not equivariant w.r.t. the symmetry group of the underlying domain. In particular, we lose the translation equivariance of convolutional layers for planar domains.

We remark that for an equidistant grid and constant[4] quadrature weights, $q = q_x$, evaluating $\mathrm{GNO}_\kappa[v]$ at $y \in D^h$ corresponds to a standard convolution as in (4) with kernel

$$K_i = q\kappa(z_i), \quad i = 1, \ldots, S,$$

where $z_i$ is defined as in (5) and $S$ is sufficiently large such that $\mathrm{supp}(\kappa) \subset [z_1, z_S]$. See Appendix C.2 for details. However, the advantage of the formulation in (1) is the fact that we can reuse the same kernel $\kappa$ across different resolutions (with the same receptive field $\mathrm{supp}(\kappa)$).

**Fourier Neural Operator (FNO).**    To retain translation equivariance, we can consider kernels of the form $k(x, y) = \kappa(x - y)$ and $U(y) = y + \mathrm{supp}(\kappa)$. Then, we can rewrite the GNO as a convolution, i.e.,

$$\mathrm{GNO}_\kappa[v] = \kappa \star v. \tag{3}$$

If we are dealing with periodic functions on the torus $D$, we can leverage the convolution theorem to compute (3), i.e., $\mathcal{F}[\kappa \star v] = \overline{\mathcal{F}[\kappa]} \cdot \mathcal{F}[v]$, where $\mathcal{F}$ maps functions to their Fourier series coefficients. The *Fourier Neural Operator* (FNO) now directly parametrizes $\overline{\mathcal{F}[\kappa]}$ and approximates $\mathcal{F}$ using the *fast Fourier transform* given that $D^h$ is an equidistant grid. While this leads to an efficient version of (2), it assumes that $\overline{\mathcal{F}[\kappa]}$ has only finitely many nonzero Fourier modes—or, equivalently, that the kernel $\kappa$ has full support, making (3) a *global* convolution.

Taking a convolutional kernel, GNOs also subsume a series of works on hypernetwork approaches for convolutional layers (Wang et al., 2018; Shocher et al., 2020).

---

[3]As long as they are suitable for the evaluation of the numerical operations of the neural operator, such as the Discrete Fourier Transform used in the Fourier neural operator, see Section 2.

[4]This is, e.g., the case for the trapezoidal rule on a torus.

**DISCO Convolutions:** We show that DISCO convolutions (Ocampo et al., 2022) represent special cases of GNOs with a convolutional kernel, which enables generalization to different geometries, and an efficient implementation, since the kernel can be pre-computed (see Section 2). In particular, DISCO convolutions are typically implemented as learnable linear combinations of fixed basis functions instead of neural networks. Moreover, they satisfy the equivariance in the limit of exact integration for special function classes.

**Scale-Equivariant CNNs:** In the area of computer vision, there has been a series of adaptations of CNNs to be (locally) scale-equivariant by using filter dilation, filter rescalings in the discrete domain (Rahman & Yeh, 2023; Sosnovik et al., 2021; Worrall & Welling, 2019) or the continuous domain (Xu et al., 2014; Sosnovik et al., 2019; Ghosh & Gupta, 2019), or input rescalings (Marcos et al., 2018). Moreover, filter rescaling has also been explored for more general group-convolutions (Bekkers, 2019).

**Neural operators and convolutional layers:** While neural operators have been developed independently of the above approaches, one can leverage similar ideas. In particular, one can rescale (i.e., up- and downsample) input and output functions of a convolutional layer or the kernel itself to obtain neural operators. We note that these approaches are, in principle, also applicable to a trained network (without the need for retraining).

In combination with FNOs, interpolation of the input and output functions has been explored by Wen et al. (2022). Also Ye et al. (2023; 2022) emphasized the need for local convolutions in FNOs, however, they do not discuss the application to different discretizations. Combined with a correct treatment of the bandlimit of the functions (based on Karras et al. (2021)), the convolutional neural operator Raonić et al. (2023) also uses up- and downsampling of the input and output functions to apply a U-Net architecture.

We note that for bandlimited functions sampled above the Nyquist frequency, the discrete convolutions have a unique correspondence to convolutions in function space, representing a special case of the DISCO framework. However, downsampling the input function introduces error for many cases where the function is not bandlimited.

To summarize, current neural operator architectures suffer from at least one of the following limitations (see Table 1): (1) they cannot succinctly represent operations with a local receptive field, e.g., FNO, or (2) they cannot be applied to different resolutions without relying on explicit up-/downsampling which may degrade performance, e.g., CNO, or (3) they cannot be scaled to obtain sufficient expressivity, since they incur prohibitively high computationally costs, such as, e.g., GNOs.

## A.1 MOTIVATION: CONVOLUTIONAL LAYER

We take inspiration from convolutional layers since they represent the prototypical version of an efficient, local operation in neural networks. However, we will see that they are not consistent in function spaces; in particular, they converge to a pointwise linear operator when we refine the discretization of the input function $v$.

Let us start by recalling the definition of a convolutional[5] layer, specifically a stride-1 convolution with $n$ input channels, a single[6] output channel, and kernel $K = (K_i)_{i=1}^S \subset \mathbb{R}^n$ of (odd) size $S$. Assuming a regular grid, i.e., $D^h = \{x_j\}_{j=1}^m \subset \mathbb{R}$ with $x_{j+1} - x_j = h$, we can define the output of the convolutional layer at $y \in D^h$ as

$$\text{Conv}_K[v](y) = (K \star \{v(x_j)\}_{j=1}^m)(y)$$

$$= \sum_{i=1}^S K_i \cdot v(z_i + y), \tag{4}$$

with

$$z_i = h\left(i - 1 - \frac{S-1}{2}\right), \tag{5}$$

---

[5]In line with deep learning frameworks, we consider convolution with the reflected filter, also known as *cross-correlation*.

[6]This is for notational convenience; the extension to multiple output channels is straightforward.

where we use zero-padding, i.e., $v(x) = 0$ for $x \notin D$.

If we now take the same kernel $K$ for finer discretizations, i.e., $h \to 0$, we see from (5) that $z_i \to 0$ and therefore,

$$\lim_{h \to 0} \mathrm{Conv}_K[v](y) = \bar{K} \cdot v(y) \quad \text{with} \quad \bar{K} = \sum_{i=1}^{S} K_i,$$

given that the function $v$ is continuous at $y$. In other words, the receptive field with respect to the underlying domain $D$ is shrinking to a point, and the convolutional layer is converging to a *pointwise linear operator*. One way to circumvent this issue would be to downsample the function $v$ appropriately to a pre-defined grid. This is done for previous approaches mentioned in Appendix A, at the cost of losing high-frequency information in the input.

In the following, we will present two ways of working on different input resolutions, while not collapsing to a pointwise operator, see also Figure 1. First, we show that rescaling (4) by the reciprocal resolution $\frac{1}{h}$ and constraining the kernel $K$ leads to *differential operators*. Then, we define the kernel $K$ as the evaluation of a function over a fixed input domain, leading to *integral operators*.

## B    GENERAL DIFFERENTIAL KERNELS

In the following, we present the general idea for irregular grids, from which Proposition 2.1 follows as a special case. Let us first define the assumptions on our grids.

**Regular discrete refinement:**    Let $\| \cdot \|$ be a norm on $\mathbb{R}^d$ and denote by $B_h(x) \subset \mathbb{R}^d$ the ball of radius $h \in (0, \infty)$ around $x \in \mathbb{R}^d$ w.r.t. $\| \cdot \|$. Further, let $D \subset \mathbb{R}^d$ be a domain and let $(h_\ell)_{\ell \in \mathbb{Z}}$ be a sequence that converges to zero. We call $(D_\ell)_{\ell \in \mathbb{Z}} \subset D$ a *regular discrete refinement* with widths $(h_\ell)_{\ell \in \mathbb{Z}}$ if there exists $N \in \mathbb{N}$ such that for all $\ell \in \mathbb{Z}$ and $x \in \mathbb{R}^d$ we have that

$$|B_{h_\ell}(x) \cap D_\ell| \leq N \quad \text{and} \quad \mathrm{span}\left(\left\{ \begin{bmatrix} 1 \\ y \end{bmatrix} - \begin{bmatrix} 1 \\ x \end{bmatrix} : y \in B_{h_\ell}(x) \cap D_\ell \right\}\right) = \mathbb{R}^{d+1}.$$

The second assumption states that we can find an *affinely independent* subset in each ball. Note that, for instance, equidistant grids satisfy these assumptions.

**First-order differential operator:**    We want to find bounded kernels $k(x, y) \colon \mathbb{R}^d \times \mathbb{R}^d \to \mathbb{R}$ with the following property: There exist $c \in \mathbb{R}$ and $b \in \mathbb{R}^d$ such that for all $v \in C^1(D, \mathbb{R})$, all $y \in \mathbb{R}^d$, and any regular discrete refinement $(D_\ell)_{\ell \in \mathbb{Z}} \subset \mathbb{R}^d$ with widths $(h_\ell)_{\ell \in \mathbb{Z}}$ it holds that

$$\lim_{\ell \to \infty} \sum_{x \in B_{h_\ell}(y) \cap D_\ell} k(x, y) v(x) = cv(y) + \nabla v(y) \cdot b. \tag{6}$$

We will now investigate which additional assumptions are needed. Let us fix $y \in \mathbb{R}^d$ and enumerate

$$(x_j)_{j=1}^{m} := B_{h_\ell}(y) \cap D_\ell.$$

Then, we can use Taylor's theorem to show that

$$\sum_{x \in B_{h_\ell}(y) \cap D_\ell} k(x, y) v(x) = \sum_{j=1}^{m} k(x_j, y) \left(v(y) + \nabla v(y) \cdot (x_j - y) + \mathcal{O}(h_\ell)\right) \tag{7}$$

$$= v(y) \sum_{j=1}^{m} k(x_j, y) + \nabla v(y) \cdot \left(\sum_{j=1}^{m} k(x_j, y)(x_j - y)\right) + \mathcal{O}(m\|k\|_{L^\infty} h_\ell). \tag{8}$$

Since the kernel and $m$ are bounded (uniformly over $\ell \in \mathbb{N}$), we have that $\mathcal{O}(m\|k\|_{L^\infty} h_\ell) = \mathcal{O}(h_\ell)$. As we want to guarantee convergence to $cv(y) + \nabla v(y) \cdot b$ for suitable $c \in \mathbb{R}$ and $b \in \mathbb{R}^d$ (independent of $y$), we need to satisfy that

$$c = \sum_{j=1}^{m} k(x_j, y) \tag{9}$$

and that

$$b = \sum_{j=1}^{m} k(x_j, y)(x_j - y) \tag{10}$$

This yields the linear[7] system

$$\underbrace{\begin{bmatrix} 1 & \dots & 1 \\ x_1 - y & \dots & x_m - y \end{bmatrix}}_{\in \mathbb{R}^{(d+1) \times n}} \begin{bmatrix} k(x_1, y) \\ \vdots \\ k(x_m, y) \end{bmatrix} = \begin{bmatrix} c \\ b \end{bmatrix}. \tag{11}$$

Our assumptions on the refinement guarantee that we can find linearly independent columns such that we can solve the system. However, generally and by abuse of notation, the value of $k(x_j, y)$ depends on all the points $(x_j)_{j=1}^{m}$. However, for an equidistant grid, one can directly see that the convolutional kernel given in Proposition 2.1 satisfies (11) with $c = 0$.

*Remark* B.1 (Higher-order differential operator). Similar to Proposition 2.1, we could approximate $k$-th order differential operator with further constraints on the elements of $K$ and a scaling factor of $\frac{1}{h^k}$. However, we do not implement them in practice, since we can also approximate higher-order derivatives by composing first-order differential layers.

## C   DISCRETE-CONTINUOUS CONVOLUTIONS

The following section outlines the basic ideas behind discrete-continuous convolutions as introduced by Ocampo et al. (2022). To generalize the (continuous) convolution in (3) to Lie groups and quotient spaces of Lie Groups, we consider the group convolution (see, e.g., Cohen & Welling (2016)).

### C.1   GENERAL DISCRETE-CONTINUOUS CONVOLUTIONS

The local support of the kernel in (1), allows us to efficiently evaluate local convolutions in subdomains of $\mathbb{R}^d$ using sparse matrix-vector products. However, the operation $y - x$, which shifts the convolution kernel is not well-defined on manifolds such as the sphere. To generalize the previous discussion to a Lie group $G$, we first replace the shift operator with the group action $g$ and obtain the so-called group convolution.

**Definition C.1** (Group Convolution). Let $\kappa, v : G \to \mathbb{R}$ be functions defined on the group $G$. The group convolution is given by

$$(\kappa \star v)(g) = \int \kappa(g^{-1}x) \cdot v(x) \, d\mu(x), \tag{12}$$

where $g, x \in G$ and $d\mu(x)$ is the invariant Haar measure on G.

*Remark* C.2. In some cases, signals are not defined on a group but rather on a quotient space $G/H$, where $H$ is a subgroup of $G$. In such cases, a convolution may still be defined by taking $g \in G/H$. For an example, see spherical convolutions (Driscoll & Healy, 1994; Ocampo et al., 2022).

This formulation presents us with the challenge of being non-trivial to discretize. While group convolutions can typically be computed by generalized Fourier transforms on the corresponding manifolds, their usage is generally preferred if the convolutions are non-local operators, i.e. the convolution kernel $k$ is not compactly supported. On the periodic domain $\mathbb{T}^n$ (i.e. Euclidean space with periodic boundaries), convolutions are typically computed discretely by directly sliding the kernel. The framework of DISCO convolutions (Ocampo et al., 2022) achieves this by approximating the integral with a quadrature rule, while evaluating the group action continuously:

**Definition C.3** (DISCO convolutions). Given a quadrature rule with quadrature points $x_j \in G$ and quadrature weights $q_j$, we approximate the group convolution (12) with the discrete sum

$$(\kappa \star v)(g) = \int \kappa(g^{-1}x) \cdot v(x) \, d\mu(x) \approx \sum_{j=1}^{m} \kappa(g^{-1}x_j) \cdot v(x_j) \, q_j. \tag{13}$$

In particular, the group action $g$ is applied analytically to the kernel function $\kappa$, whereas the integral is approximated using the quadrature rule.

---

[7]For fixed $y$.

For a discrete set of output locations $g_i$, this becomes a straight-forward matrix-vector multiplication

$$\sum_{j=1}^{m} \kappa(g_i^{-1} x_j) \cdot v(x_j)\, q_j = \sum_{j=1}^{m} K_{ij} \cdot v(x_j)\, q_j \qquad (14)$$

with $K_{ij} = \kappa(g_i^{-1} x_j)$. In the case where $\kappa$ is compactly supported, $K_{ij}$ is a sparse matrix with the number of non-zero entries per row depending on the resolution of the grid $x_j$ and the support of $\kappa$. To obtain a learnable filter, $\kappa$ is parametrized as a linear combination of a chosen set of basis functions.

Although we have presented the general idea only for Lie groups $G$, it is possible to construct the DISCO convolution also on manifolds with a group action acting on them, such as the 2-sphere $\mathbb{S}^2$. For a detailed discussion of DISCO convolutions and a construction in one dimension, we point the reader to Ocampo et al. (2022).

*Remark* C.4 (Exact integration and equivariance). We note that DISCO convolutions satisfy equivariance properties for function classes which can be exactly integrated. For instance, on a an equidistant grid, these could be polynomials when using Legendre points. On the torus and equidistant grids, exact integration holds for bandlimited functions that are sampled above the Nyquist frequency.

### C.2 DISCO CONVOLUTIONS IN ONE DIMENSION

For the sake of simplicity, we discuss the simple one-dimensional case on $D = [0, 1]$ with periodic boundary conditions. We note that this corresponds to the circle group (or the torus) $\mathbb{T}$. For any element $y \in D$, the corresponding group operation $T_y$ is the translation $T_y : D \to D,\, x \mapsto x \oplus y$, where we denote by $x \oplus y$ a modular shift such that the result remains in $D$.

Then, the DISCO convolution in one dimension, for Lebesgue square-integrable functions $v$ and $\kappa$ becomes

$$(\kappa \star v)(y) = \int_{[0,1]} \kappa(T_y^{-1} x) v(x)\, \mathrm{d}x = \int_{[0,1]} \kappa(x - y) v(x)\, \mathrm{d}x \approx \sum_{j=1}^{m} \kappa(x_j - y)\, v(x_j)\, q_j, \quad (15)$$

for suitable quadrature points $D^h = \{x_j\}_{j=1}^{m}$ with corresponding quadrature weights $\{q_j\}_{j=1}^{m}$.

To parameterize the filter $\kappa$, we pick a finite support $[0, x_{\text{cutoff}}]$ with $L$ equidistant collocation points $\xi^{(\ell)} \in [0, x_{\text{cutoff}}]$ and corresponding hat functions. The $\ell$-th hat function is then defined as

$$\kappa^{(\ell)}(x) = \begin{cases} \frac{x - \xi^{(\ell-1)}}{\xi^{(\ell)} - \xi^{(\ell-1)}} & \text{for } \xi^{(\ell-1)} \le x < \xi^{(\ell)} \\ \frac{\xi^{(\ell+1)} - x}{\xi^{(\ell+1)} - \xi^{(\ell)}} & \text{for } \xi^{(\ell)} \le x < \xi^{(\ell+1)} \\ 0 & \text{else,} \end{cases} \qquad (16)$$

where $\xi^{(0)}, \xi^{(L+1)} \in [0, x_{\text{cutoff}}]$ are suitable boundary points. The resulting filter is obtained as a linear combination $\kappa = \sum_{\ell=1}^{L} \theta^{(\ell)} k^{(\ell)}$ with trainable parameters $\theta^{(\ell)}$. Plugging this into (14), we obtain the trainable DISCO convolution

$$\sum_{j=1}^{m} \kappa(x_j - y_i)\, v(x_j)\, q_j = \sum_{\ell=1}^{L} \sum_{j=1}^{m} \theta^{(\ell)}\, \kappa^{(\ell)}(x_j - y_i)\, v(x_j)\, q_j = \sum_{\ell=1}^{L} \sum_{j=1}^{m} \theta^{(\ell)} K_{ij}^{(\ell)}\, v(x_j)\, q_j, \quad (17)$$

where $K_{ij}^{(\ell)} = \kappa^{(\ell)}(x_j - y_i)$ are the shifted filter functions.

Let us now consider the special case of an equidistant grid $D^h$, i.e., $x_{j+1} - x_j = h$, and a trapezoidal quadrature rule $q_j = h$. Let us further assume that the output points $y_i$ coincide with the grid points and that the collocation points $\xi^{(\ell)}$ are given as the first $L$ grid points. Then, due to the property of the hat functions, $K_{ij}^{(\ell)}$ can only contain either $0$ or $1$ and we obtain the circulant convolution matrices given by

$$K^{(1)} = (e_1, e_2, e_3, \ldots, e_m), \quad K^{(2)} = (e_2, e_3, \ldots, e_m, e_1), \quad K^{(3)} = (e_3, \ldots, e_m, e_1, e_2), \quad \ldots$$

where $e_i \in \mathbb{R}^m$ is the $i$-th standard basis vector. For regular grids on planar geometries, we can thus efficiently implement the DISCO convolution in (17) in terms of highly-optimized CUDA kernels based on common-place convolutional layers. This observation also generalizes to higher dimensions, where the same discrete kernel is obtained when the kernel is continuously shifted on the grid.

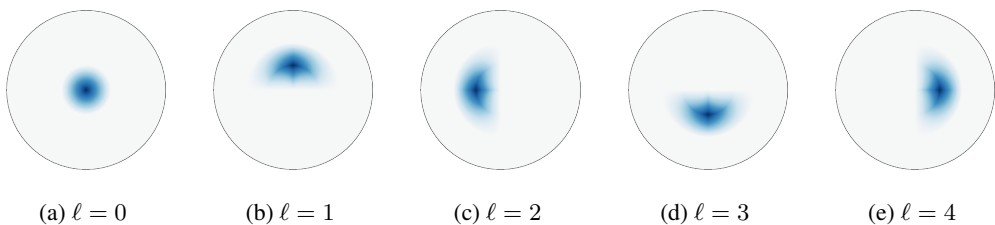

(a) $\ell = 0$     (b) $\ell = 1$     (c) $\ell = 2$     (d) $\ell = 3$     (e) $\ell = 4$

Figure 3: Radial, piecewise linear basis functions for the approximation of anisotropic filters on the sphere.

### C.3 DISCO CONVOLUTIONS ON THE SPHERE

For general group actions $g \in SO(3)$, the outcome of the group convolution (12) will be a function defined on $SO(3)$. This is due to $\mathbb{S}^2$ not being a group, but rather a manifold on which $SO(3)$ acts. We can see this by fixing the north pole $n = [0, 0, 1]^\top$ and applying any rotation $g \in SO(3)$ to it. This will trace out the whole sphere, despite the north-pole eliminating one of the Eulerian rotation angles. Therefore, to ensure that the result of the convolution is still a function defined on $\mathbb{S}^2$, we can simply restrict $g$ in (12) to rotations in $SO(3)/SO(2)$, which is isomorphic to $\mathbb{S}^2$. Formally, this can be achieved by fixing the first of the three Euler angles parameterizing $g$ to 0.

As basis functions we pick a set of piecewise linear basis functions as in (16). To accomodate anisotropic kernels, collocation points are distributed in an equidistant manner along both radius and circumference. More precisely, the first collocation point lies at the center, and for each consecutive ring, a fixed amount of collocation points is distributed along the circumference. The resulting basis functions are illustrated in Figure 3, for a cutoff radius of $r_{\text{cutoff}} = 0.1\pi$.

## D DATASETS

In this section, we outline the datasets for our numerical experiments.

### D.1 DARCY FLOW

We first consider the steady-state, two-dimensional Darcy Flow equation

$$-\nabla \cdot (a\nabla u) = f, \quad u|_{\partial D} = 0, \tag{18}$$

on the domain $D = (0, 1)^2$. In this problem, we motivate the need for incorporating local operators as inductive biases in neural operator architectures. To this end, we explicitly construct a problem that requires approximating the forcing function $f$ for a given diffusion coefficient $a$ and pressure $u$. In other words, the task consists of learning the differential operator

$$u \mapsto -\nabla \cdot (a\nabla u). \tag{19}$$

Following the setup of Hasani & Ward (2024), we take

$$a(x) = \begin{bmatrix} x_1^2 & \sin(x_1 x_2) \\ x_1 + x_2 & x_2 \end{bmatrix} \tag{20}$$

and

$$u(x) = \sum_{i,j=1}^{20} \frac{c_{ij}}{\sqrt{\lambda_{ij}}} \sin(i\pi x_1) \sin(j\pi x_2), \tag{21}$$

for $x \in D$, where,

$$\lambda_{ij} = (i\pi)^2 + (j\pi)^2$$
$$c_{ij} \sim \mathcal{N}(0, 1/(i+j))$$

We then compare the performance of our proposed models: FNO in parallel with differential kernels, FNO with integral kernels, and FNO with both kernels. We also compare with a baseline FNO and the U-Net architecture from Gupta & Brandstetter (2022). Moreover, we perform zero-shot super-resolution on this problem; these results can be found in Appendix E.4.

## D.2 NAVIER-STOKES EQUATIONS

Next, we consider the two-dimensional Navier-Stokes equation on the torus. In particular, we consider Kolmogorov flows, which can be described by

$$\partial_t u + u \cdot \nabla u - \frac{1}{\text{Re}} \Delta u = -\nabla p + \sin(ny)\hat{x}$$
$$\nabla \cdot u = 0 \tag{22}$$

on the spatial domain $D = [0, 2\pi]^2$ equipped with periodic boundary conditions. In the above, $u$ and $p$ denote the velocity and pressure, and Re denotes the Reynolds number of the flow. We want to learn the solution operator mapping initial conditions $u(\cdot, 0) = u_0$ to the time-evolved solution in vorticity form $w(\cdot, \tau)$ at time $\tau \in (0, \infty)$, where the vorticity is given by

$$w = (\nabla \times u)\hat{z}, \tag{23}$$

with $\hat{z}$ being a unit vector normal to the plane.

We use the dataset from Li et al. (2022), which sets $m = 4$ and Re $= 5000$ in (22) and uses a temporal discretization of $\tau = 1$ on a $128 \times 128$ regular grid. The initial conditions are sampled from a Gaussian measure as described in Li et al. (2022), and the equation is solved with a pseudo-spectral solver. We emphasize that learning the solution operator for such a high Reynolds number is very challenging due to the turbulent nature and small-scale features of the flow. We conjecture that the baseline FNO is prone to over-smoothing over the finer scales, consequently leading to a degradation of performance.

To validate this conjecture, we compare the performance of the local neural operator with a baseline FNO and U-Net and evaluate the effectiveness of our proposed differential and integral kernels, respectively.

## D.3 SHALLOW WATER EQUATIONS

Finally, we test our proposed architecture on the shallow water equations on the rotating sphere. They represent a system of hyperbolic partial differential equations used to model a variety of geophysical flow phenomena, such as atmospheric flows, tsunamis, and storm surges. They can be formulated in terms of the evolution of two state variables $\varphi$ and $u$ (geopotential height and the tangential velocity of the fluid column), governed by the equations

$$\partial_t \varphi + \nabla \cdot (\varphi u) = 0,$$
$$\partial_t (\varphi u) + \nabla \cdot F = f, \tag{24}$$

on the sphere $D = \mathbb{S}^2$ with suitable initial conditions $\varphi(\cdot, 0) = \varphi_0$ and $u(\cdot, 0) = u_0$, a momentum flux tensor

$$F_{ij} = \varphi u_i u_j + \frac{1}{2}\varphi^2, \tag{25}$$

and a source term

$$f(x) = -2\Omega x \times (\varphi u), \quad x \in \mathbb{S}^2, \tag{26}$$

which models the Coriolis force due to the rotation of the sphere with angular velocity $\Omega$. We use the dataset from Bonev et al. (2023), which uses a Gaussian random field to generate initial conditions on the sphere at a resolution of $256 \times 512$ and solves for $\varphi$, $u$ at a lead time of one hour. The target solution is computed using a spectral solver, which takes 24 Euler steps to compute the reference[8].

As baselines we use a planar U-Net architecture, a spherical U-Net, where convolutions are performed using convolutions with local integral kernels on the sphere, and an SFNO architecture. On the sphere, we only consider the impact of the integral kernel on the SFNO architecture, as it allows for a natural extension to the sphere, see Appendix C.1.

## E EXPERIMENTAL DETAILS

A numerical comparison of our methods with baseline FNO and U-Net architectures can be found in Table 2. In this section, we outline the implementation details for our numerical experiments. In

---

[8]The dataset and solver are taken from the `torch-harmonics` package at https://github.com/NVIDIA/torch-harmonics.

Table 3: Overview over the hyperparameters selected for out different experiments and architecture.

| Model | Parameters | | | |
|---|---|---|---|---|
| | # Layers | # Modes | Embedding | # Parameters |
| Darcy Flow | | | | |
| U-Net | 17 | - | 18 | $2.850 \cdot 10^6$ |
| FNO | 4 | 20 | 41 | $2.715 \cdot 10^6$ |
| **FNO + diff. kernel (ours)** | 4 | 12 | 65 | $2.638 \cdot 10^6$ |
| FNO + local integral kernel (ours) | 4 | 20 | 40 | $2.617 \cdot 10^6$ |
| FNO + local integral + diff. kernel (ours) | 4 | 12 | 64 | $2.639 \cdot 10^6$ |
| Navier-Stokes Equations | | | | |
| U-Net | 17 | - | 56 | $2.758 \cdot 10^7$ |
| FNO | 4 | 40 | 65 | $2.711 \cdot 10^7$ |
| FNO + diff. kernel (ours) | 4 | 40 | 65 | $2.726 \cdot 10^7$ |
| FNO + local integral kernel (ours) | 4 | 20 | 129 | $2.716 \cdot 10^7$ |
| **FNO + local integral + diff. kernel (ours)** | 4 | 20 | 127 | $2.691 \cdot 10^7$ |
| Spherical Shallow Water Equations | | | | |
| U-Net | 17 | - | 32 | $2.898 \cdot 10^6$ |
| Spherical U-Net (with local integral kernel) | 17 | - | 32 | $1.639 \cdot 10^6$ |
| SFNO | 4 | 128 | 32 | $1.066 \cdot 10^6$ |
| **SFNO + local integral kernel (ours)** | 4 | 128 | 31 | $1.019 \cdot 10^6$ |

all three problems, training is conducted by minimizing the squared $L^2$-loss until convergence is reached. Our U-Net baseline is adapted from the model and code of the PDE Arena benchmark Gupta & Brandstetter (2022). For the FNOs Li et al. (2020a) and SFNOs Bonev et al. (2023), we use the implementation in the `neuraloperator` and `torchharmonics` libraries. Moreover, for all experiments, GELU activation functions and the Adam optimizer are used.

For all three problems, we trained the FNO/SFNO-based models with varying widths and modes, while keeping the overall number of parameters approximately constant. We present the best results for each problem and macro-architecture in Table 2. For the models with local layers, we found that a larger width and fewer modes can often improve performance. We conjecture that the increased embedding dimension confers additional expressivity to the local kernels. This also suggests that local operators are an important inductive bias for these problems. A detailed experimental setup is outlined for all three datasets in the following subsections.

### E.1 2D DARCY FLOW EQUATION

In the 2D Darcy flow setting, we generate our data as described in Appendix D.1. We train and test our models and baselines with data discretized onto a $256 \times 256$ regular grid. As a baseline, we compare our proposed models with FNO Li et al. (2020a) and the U-Net architecture of Gupta & Brandstetter (2022). We note that this U-Net architecture is not agnostic to the discretization Kovachki et al. (2021), see also Appendix A.1. We compare these baselines against our proposed models: the architecture using convolutions with Fourier (i.e., FNO), differential, and integral kernels, as well as architectures using only FNO and differential kernels or only FNO with the proposed integral kernels. Figure 4 compares the predictions of each model.

We choose all hyperparameters such that the overall number of parameters of all compared models is similar. The number of layers, number of Fourier modes, and embedding dimension are shown in Table 2. Models using convolutions with differential and local integral kernels use these layers in parallel to the Fourier layers and pointwise skip connection on all layers. We use reflective padding for the convolutional operations in the differential and local integral kernels. For all relevant models, the local integral kernels use a radius cutoff of $0.007$ on $[-1, 1]^2$, and they are parameterized by five radial, piecewise linear basis functions for the approximation of anisotropic filters (analogous to Figure 3 on the plane). The differential kernels are parameterized as $3 \times 3$ convolutional kernels over the regular grid. For our U-Net baseline, we use $3 \times 3$ convolutional kernels with 2 residual

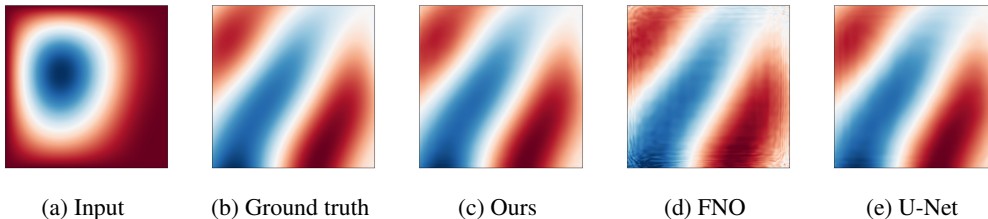

| (a) Input | (b) Ground truth | (c) Ours | (d) FNO | (e) U-Net |

Figure 4: Comparison of models from Table 2: The outputs of our best-performing model, FNO, and U-Net on a randomly-selected input pressure function from the Darcy flow problem. Edge artifacts are very prevalent on the FNO predictions, and they are less dominant in the U-Net predictions.

blocks for each resolution (two downsampling and two upsampling) and three layers within each block, with channel multipliers of $1, 2, 4$ for each layer within a block.

Training is conducted by minimizing the squared $L^2$-loss for 70 epochs on a single NVIDIA P100 GPU, which is sufficient to achieve convergence on all models. We used a step learning rate decay scheduler, starting at $10^{-3}$ and halving every 10 epochs. The results are shown in Table 2.

### E.2    2D NAVIER-STOKES EQUATIONS

For the 2D Navier-Stokes equations, we use the same experimental setup and dataset as Li et al. (2022). The data is discretized on a $128 \times 128$ regular grid. The particular form of the Navier-Stokes equation that we consider is a 2D Kolmogorov flow with a Reynolds number of 5000. We compare the same five models as in the Darcy experiment.

As in the Darcy setting, we choose all hyperparameters such that the overall number of parameters for all the models is similar. The number of layers, number of Fourier modes, and embedding dimension are shown in Table 2. Models using convolutions with differential and local integral kernels use these layers in parallel to the Fourier layers and pointwise skip connection on all layers. We enforce periodic boundary conditions during padding for the convolutional operations in the differential and local integral kernels. For all relevant models, the local integral kernels use a radius cutoff of $0.05\pi$ on the torus, and they are parameterized by five radial, piecewise linear basis functions for the approximation of anisotropic filters (analogous to Figure 3 on the plane). The differential kernels are parameterized as $3 \times 3$ convolutional kernels over the regular grid. Our U-Net baseline is set up in the same way as in the Darcy experiment.

Training was performed for 136 epochs on a single NVIDIA RTX 4090 GPU with an exponentially decaying learning rate, starting at $10^{-3}$ and halving every 33 epochs. The results are shown in Table 2.

### E.3    SHALLOW WATER EQUATIONS

For the shallow water equations, we use the dataset presented by Bonev et al. (2023). To generate samples, initial conditions are drawn from a Gaussian random process at a resolution of $256 \times 512$ on an equiangular lat-lon-grid and then advanced in time using a numerical spectral solver. Physical constants such as the sphere's radius or the Coriolis force are set to match those of Earth. The numerical solver uses 150 explicit Euler steps to advance the solution 1 hour in time. The right-hand side is discretized using the spectral basis provided by the Spherical Harmonics. For a detailed description of the dataset, we refer the reader to Bonev et al. (2023) and the corresponding implementation in the `torch-harmonics` package.

As a baseline for our experiments, we use the SFNO architecture as presented by Bonev et al. (2023), where the embedding dimension is adjusted to 32 to obtain a manageable parameter count. Moreover, we adapt the U-Net architecture by Gupta & Brandstetter (2022) to the spherical domain by replacing all spatial (i.e., not the $1 \times 1$) convolutions with DISCO convolutions on the sphere. Therefore, the resulting architecture is a spherical U-Net similar to the architecture presented by Ocampo et al. (2022). Moreover, due to the DISCO convolutions' discretization-agnostic nature, this architecture can be interpreted as a neural operator. Finally, we augment the SFNO architecture with local DISCO convolutions to obtain the proposed architecture; see Section 2.

Table 4: Zero-shot super-resolution results for Darcy flow and the spherical shallow water problems. The validation error is reported in terms of relative $L^2$ loss at double the training resolution. Autoregressive rollouts are super-resolved in the sense that the rollout is performed at the high resolution.

| Model | Relative $L^2$ Error. (super-res.) | |
| --- | --- | --- |
| | 1 step | 5 steps |
| Darcy Flow | | |
| FNO | $8.646 \cdot 10^{-2}$ | - |
| **FNO + diff. conv. (ours)** | $\mathbf{7.774 \cdot 10^{-2}}$ | - |
| Spherical Shallow Water Equations | | |
| Spherical U-Net (with local integral conv.) | $3.386 \cdot 10^{-3}$ | $1.244 \cdot 10^{-2}$ |
| SFNO | $3.830 \cdot 10^{-3}$ | $1.427 \cdot 10^{-2}$ |
| **SFNO + local integral conv. (ours)** | $\mathbf{3.232 \cdot 10^{-3}}$ | $\mathbf{7.450 \cdot 10^{-3}}$ |

For all three architectures, hyperparameters were chosen to achieve roughly similar parameter counts. The learning rates for each architecture were determined with a quick parameter sweep, resulting in $3 \cdot 10^{-4}$ for the spherical U-Net and $3 \cdot 10^{-3}$ for both neural operators. As a learning rate scheduler, we use the policy of halving the learning rate upon a plateauing of the loss. Training was performed for 100 epochs on a single NVIDIA RTX A6000 GPU, which was sufficient to achieve convergence on all considered models.

### E.4    ZERO-SHOT SUPER-RESOLUTION RESULTS

In this paper, we propose two methods to embed the inductive bias of locality into neural operator architectures. The key distinction between our proposed methods and CNN-based architectures is that our methods are agnostic to the discretization of the input function. In this section, we present and discuss the super-resolution capabilities of our proposed models. In particular, we focus on two examples: (1) the Darcy flow equation to showcase our differential layers on a regular Cartesian grid and (2) the shallow water equation to demonstrate super-resolution for our local integration on the sphere. The experimental setting that we consider is that of *zero-shot super-resolution*. In particular, suppose that the model has been trained at a particular training resolution (or at multiple different resolutions). Given input at a higher resolution than the training resolution, the task of zero-shot super-resolution is to then predict the output function at this higher resolution and evaluate the resulting model error.

In the Darcy flow setting, we use the same setup and dataset as described in Appendix D.1. We train two models on data sampled at a $256 \times 256$ regular grid and evaluate on a $512 \times 512$ regular grid (2x super-resolution and 4x the number of points). For the shallow water equations, a similar approach is taken. We take the models in Appendix D.3 which were trained at a resolution of $256 \times 512$ and apply them to data generated at a resolution of $512 \times 1024$.

Table 4 lists the results of our super-resolution experiment with our proposed models. In the Darcy setting, we compare the baseline FNO to the FNO augmented with the differential kernel. On the sphere, we make use of the fact that the spherical U-Net presented in Ocampo et al. (2022) is already a neural operator due to its discretization-independence. As such, we can use it alongside the SFNO as a baseline for zero-shot super-resolution on the sphere.

As with all neural operator architectures, during training there is the possibility of overfitting to the training resolution. For instance, FNO may learn features in Fourier space that are intrinsically tied to the resolution of the input function. Similarly, it is possible that our proposed differential and integral convolutional operators will learn a function of the training discretization. This effect can be remedied by using high-resolution training data where the local details are fully resolved. This minimum required resolution of the training data is thus a function of the smoothness of the input function. For this reason, we decided to exclude the 2D Navier-Stokes problem from our super-resolution experiments, since training at a resolution that sufficiently resolves the local dynamics would be prohibitvely expensive.

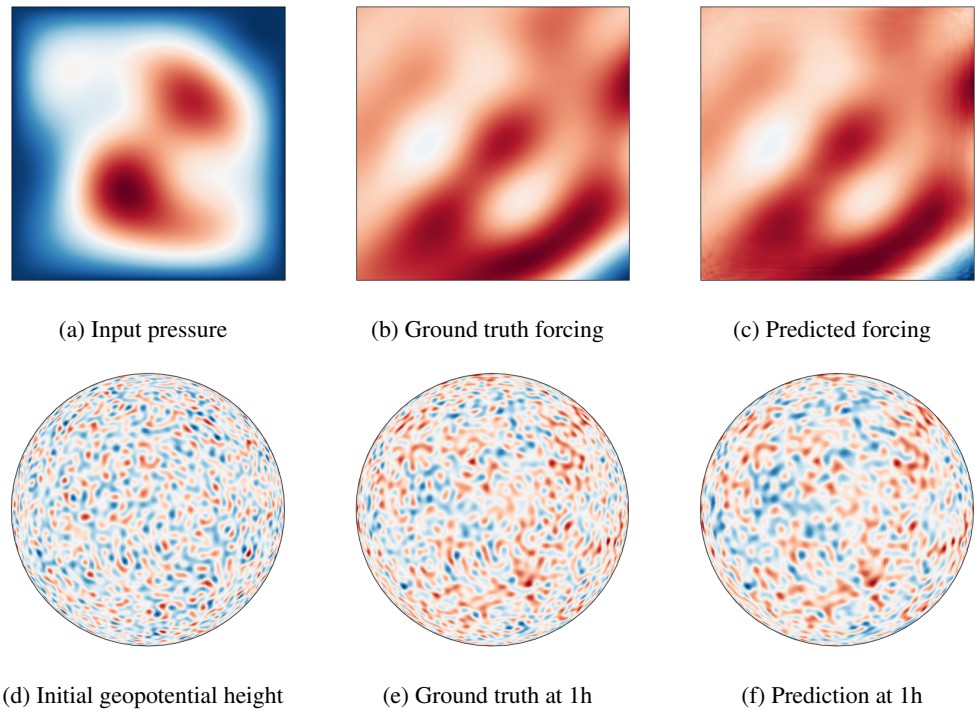

Figure 5: Randomly-selected super-resolution samples for the Darcy flow (top row) and shallow water (bottom row) problems.

In our experiments, we noticed that our differential layers tend to incur some discretization errors when trained on data that is not of sufficiently high resolution. If differential convolutions are present in consecutive layers in the model, this error can propagate quickly. As such, we found that the best-performing model on zero-shot super-resolution for the Darcy flow problem is a model with a differential convolution in only the first layer. Using fewer differential layers in our model reduces the expressivity, but we find that the super-resolution capabilities are still better than the baseline FNO.

Lastly, we would like to note that in our experiments, FNO had a larger error near the boundary in non-periodic problems, as the FFT used in FNO assumes periodic boundary conditions. We note that our proposed differential layer can help reduce the error at the boundary caused by FNO, but some of these effects may still be present (see Figure 5).

