# OpenReview forum: "Neural operators with localized integral and differential kernels"
_ICLR.cc/2024/Workshop/AI4DiffEqtnsInSci — AI4DiffEqtnsInSci @ ICLR 2024 Oral_

### Official Review · Reviewer_2NZe · 2024-02-23
**A novel perspective on achieving spatially localized feature learning in NOs**

**Rating:** 8
**Confidence:** 3

**Review:**

This work introduces the concepts of local and differential kernels for neural operator architectures. This is to help address a specific limitation of spectral neural operators, in that they only perform a global convolution in Fourier space.

I think the ideas are quite interesting, and the results suggest the techniques work well to improve FNO and SFNO. I appreciate the detailed discussion about related approaches to achieve a similar result in Appendix A Related Work. A minor suggestion to improve the clarity of the work would be to summarize in plain words what exactly goes into the local differential kernel (e.g., "we center the kernel by subtracting its mean and re-scaling the result by the reciprocal resolution) and integral kernel early on, perhaps in Figure 1's caption. Another would be to include a detailed Limitations section.

---

### Official Review · Reviewer_Xy67 · 2024-02-24

**Rating:** 9
**Confidence:** 5

**Review:**

This paper proposes two local operations using convolutions with a differential kernel and a local integral kernel that can capture local features of the problem while preserving the grid resolution invariance property of neural operators. The performance of FNO is enhanced by adding differential and integral layers to FNO blocks. The paper is well-written, and the authors provide sufficient empirical evidence for the proposed improvements.
1. Does Proposition 2.1 hold for a non-uniform grid with variable grid spacing? How will the scaling factor change for non-uniform grid? How is the grid spacing determined for the spherical shallow water equation? Is the grid spacing constant for all PDEs investigated in this work?
2. The authors only compare the L2 error for different problems. For the Navier-Stokes equation at high Reynolds numbers, an energy spectra comparison should be added. It would provide insight into how the proposed differential and integral layers capture high-frequency content that is not captured by FNO due to oversmoothing.
3. The authors provide a clear justification for not performing zero-shot super resolution for the Navier-Stokes equation. How important is it to have data at different grid resolutions when training FNO? If FNO is trained on single-resolution data, can it generalize well to different grid resolutions? Or does FNO only generalize well when the training dataset includes different grid resolutions?
4. Do the 5 steps in Fig. 3 correspond to prediction after 5 hours or 5 explicit Euler steps? In the problem description for the spherical shallow water equation, it is noted that 1 hour equals 150 explicit Euler steps.
5. The implementation code is not provided, making it difficult to reproduce the results in this paper.

---

### Meta-Review · Area_Chair_4ty3 · 2024-03-01

**Recommendation:** Accept (Oral)

**Metareview:**

Thanks for the detailed reviews submitted by the two reviewers. The paper addresses the importance of capturing local effects by using local kernels and provided experimental results indicating how important the local effects are to be captured for a more accurate NO model. Reviwers have raised a few points which are useful further enhance the clarity of paper and I recommend authors to take action accordingly. My decision for this paper is accept potentially for oral presentation, or if there are papers which are more deserved to be oral, this is definitely poster accept.

---

### Decision · Program_Chairs · 2024-03-02

Accept (Oral)